# Bacterial Membrane Vesicles in Pneumonia: From Mediators of Virulence to Innovative Vaccine Candidates

**DOI:** 10.3390/ijms22083858

**Published:** 2021-04-08

**Authors:** Felix Behrens, Teresa C. Funk-Hilsdorf, Wolfgang M. Kuebler, Szandor Simmons

**Affiliations:** 1Institute of Physiology, Charité—Universitätsmedizin Berlin, Charitéplatz 1, 10117 Berlin, Germany; felix.behrens@charite.de (F.B.); teresa.funk-hilsdorf@charite.de (T.C.F.-H.); szandor.simmons@charite.de (S.S.); 2Berlin Institute of Health (BIH), 10178 Berlin, Germany; 3DZHK (German Centre for Cardiovascular Research), Partner Site Berlin, 10117 Berlin, Germany; 4The Keenan Research Centre for Biomedical Science at St. Michael’s, Toronto, ON M5B 1X1, Canada; 5Departments of Surgery and Physiology, University of Toronto, Toronto, ON M5S 1A8, Canada

**Keywords:** pneumonia, lower respiratory tract infection, extracellular vesicles, outer membrane vesicles, membrane vesicles, vaccine

## Abstract

Pneumonia due to respiratory infection with most prominently bacteria, but also viruses, fungi, or parasites is the leading cause of death worldwide among all infectious disease in both adults and infants. The introduction of modern antibiotic treatment regimens and vaccine strategies has helped to lower the burden of bacterial pneumonia, yet due to the unavailability or refusal of vaccines and antimicrobials in parts of the global population, the rise of multidrug resistant pathogens, and high fatality rates even in patients treated with appropriate antibiotics pneumonia remains a global threat. As such, a better understanding of pathogen virulence on the one, and the development of innovative vaccine strategies on the other hand are once again in dire need in the perennial fight of men against microbes. Recent data show that the secretome of bacteria consists not only of soluble mediators of virulence but also to a significant proportion of extracellular vesicles—lipid bilayer-delimited particles that form integral mediators of intercellular communication. Extracellular vesicles are released from cells of all kinds of organisms, including both Gram-negative and Gram-positive bacteria in which case they are commonly termed outer membrane vesicles (OMVs) and membrane vesicles (MVs), respectively. (O)MVs can trigger inflammatory responses to specific pathogens including *S. pneumonia, P. aeruginosa,* and *L. pneumophila* and as such, mediate bacterial virulence in pneumonia by challenging the host respiratory epithelium and cellular and humoral immunity. In parallel, however, (O)MVs have recently emerged as auspicious vaccine candidates due to their natural antigenicity and favorable biochemical properties. First studies highlight the efficacy of such vaccines in animal models exposed to (O)MVs from *B. pertussis, S. pneumoniae*, *A. baumannii,* and *K. pneumoniae*. An advanced and balanced recognition of both the detrimental effects of (O)MVs and their immunogenic potential could pave the way to novel treatment strategies in pneumonia and effective preventive approaches.

## 1. Microbial Etiology of Lower Respiratory Tract Infections

Pneumonia as the most common lower respiratory tract infection is the leading cause of infection-associated death worldwide, and the fourth most common cause of death globally [1,2]. Pneumonia is commonly classified into community-acquired pneumonia (CAP) and hospital-acquired pneumonia (HAP) [3], with HAP as the most frequent health care-associated infection [4]. Bacteria traditionally form the main causative pathogens of pneumonia [5]. While this is still the case for HAP, where *Staphylococcus aureus* and *Pseudomonas aeruginosa* dominate [6], vaccinations have changed the microbial etiology in CAP. In Europe *Streptococcus pneumoniae* (Pneumococcus) and *Haemophilus influenzae* are still the leading causes [7,8], whereas in the United States respiratory viruses have become more frequently detected as possible causes of CAP as compared to bacterial pathogens [9], as summarized in Table 1. Epidemiological data suggest a U-shaped age distribution in pneumonia patients, with both younger children and elderly as main risk groups [10]. In addition, the risk of pulmonary infections is increased in patients with chronic respiratory diseases such as chronic obstructive pulmonary disease (COPD) and cystic fibrosis (CF) and in immunocompromised individuals [11,12,13,14].

Advances in antimicrobial antibiotic therapy have drastically refined the treatment of pulmonary infections [5]. Additionally, the development of vaccines against common causes of lower respiratory tract infections, including *S. pneumoniae, H. influenzae* type B, *Bordetella pertussis,* and seasonal influenza, has significantly reduced, albeit not eliminated, morbidity and mortality due to respective infections [21,22,23]. Concomitantly, new challenges have arisen for the treatment of patients with pneumonia. First, these include adaptive changes in microbial properties, like the rise of strains resistant to current vaccinations and established antibiotic regimes [24]. Second, a series of new zoonotic respiratory pathogens causing virgin soil epidemics in humans have arisen of late including swine and avian flu, and coronaviruses such as SARS, MERS, and SARS-CoV-2. Third, the pathophysiology of both acute and long-term complications of pneumonia including acute respiratory distress syndrome, sepsis, and cardiovascular complications, remain widely unknown and, therefore, at present difficult to treat or prevent [25,26,27,28,29].

Of late, extracellular membrane vesicles released from bacteria are increasingly recognized as molecular shuttles of nucleic acids, proteins, lipids, and carbohydrates involved in bacterial pathogenicity with the ability to target and (de)regulate host cells. Consequently, bacterial membrane vesicles start to receive noticeable attention for their potential role as detrimental mediators in bacterial infections [30]. In this review we summarize recent insights into the emerging role of bacterial membrane vesicles in the pathophysiology of pneumonia and its complications, and on their adoption as auspicious targets for future preventive and therapeutic approaches.

## 2. (Outer) Membrane Vesicles—Biogenesis, Characteristics, and Analytical Methods

Extracellular vesicles (EVs) are lipid bilayer-delimited particles that are spontaneously released from cells but, unlike cells, cannot replicate [31]. The release of EVs is a highly conserved mechanism in the vast majority of cells and organisms. As such, it is not limited to complex eukaryotic organisms, but equally present in bacteria, archaea, fungi, and parasites [32]. Both Gram-positive and Gram-negative bacteria produce EVs, which are referred to as membrane vesicles (MVs) and outer membrane vesicles (OMVs), respectively, based on their proposed mechanism or release [30]: while OMVs are thought to bleb from the outer membrane of Gram-negative bacteria and as such encapsulate periplasmatic content, MVs are considered to bud from the cytoplasmatic membrane of Gram-positive bacteria and accordingly, contain cytoplasmatic components [33,34]. In addition to OMVs, it has become clear that Gram-negative bacteria also release double and even triple membrane vesicles, which could be produced upon bacterial lysis or as a result of encapsulated bacteriophages respectively, even though these hypotheses need to be affirmed by appropriate testing and, thus, remain subjects of current research [34,35]. Yet, triggers and signaling pathways stimulating (O)MV release, and the (selective) transfer of cargo into the vesicles and the molecular mechanisms regulating membrane-release remain incompletely understood. In contrast to fungi and mammalian cells, the mechanism of EV formation in intracellular multivesicular bodies, which release exosomes upon fusion with the cell membrane, does not seem to play a role in bacteria [31,33,36].

(O)MVs are sized between 20 and 400 nm and contain a variety of cargo, including both cytosolic and surface proteins, nucleic acids, and virulence factors [30,32,34,37]. Notably, membrane composition of OMVs and MVs differs profoundly as a function of the releasing organism. For example, high levels of lipopolysaccharide (LPS) are omnipresent on OMVs, but non-existent on MVs [34]. OMVs were first discovered in the 1960s in *Escherichia coli* and considered as extracellular globules transporting the extracellular lipoglycopeptide consisting of LPS and polysaccharides [33,38]. In contrast, Gram-positive bacteria, and other organisms with thick cell walls, were long considered incapable of releasing EVs. This assumption led to a protracted discovery of MVs and a sustained knowledge gap regarding the mechanisms by which MVs break through the bacterial cell wall upon their release [33]. Although important aspects of (O)MV genesis thus remain incompletely understood, recent studies have begun to shed light on the biological role of (O)MVs. Specifically, (O)MVs were found to play important roles in bacterial virulence, mediate horizontal gene transfer and other forms of cell-to-cell communication, and to confer immunomodulatory effects in host organisms [34].

Due to considerable heterogeneity in terminology, characterization, and analysis of EVs in initial studies—including those on (O)MVs—rigorous efforts have been undertaken of late with the aim of methodological standardization, and resulted in recent guidelines by the International Society for Extracellular Vesicles [39]. A broad variety of methods are presently used for both purification and isolation of EVs and in analytical approaches. Purification can be achieved by either (ultra-) centrifugation, size exclusion chromatography, affinity-based approaches, and other, less frequently applied methods, limiting direct comparability of purified EV samples in analytical and functional assays [40]. Key methods for the characterization of EVs/(O)MVs comprise (i) electron microscopy, which yields information on EV/(O)MV shape and size, yet is poorly suitable for high-throughput analysis [41]; (ii) light scattering-based methods, e.g., nanoparticle tracking analysis that allows for determination of EV/(O)MV size distribution and concentration yet without direct visualization [42]; and (iii) approaches to identify the molecular composition of EV/(O)MV. Among the latter, flow cytometry is most commonly used for the characterization of surface molecules [43], while classical biochemical and molecular biological methods and in particular innovative OMICS approaches can deliver in-depth analysis of important classes of EV/(O)MV cargo including proteins, nucleic acids, and lipids [39,44,45,46]. As none of these methods provides a comprehensive characterization of EVs/(O)MVs on its own, a panel of methods is ideally employed in order to provide robust data on EVs’/(O)MVs’ physicochemical and biochemical properties.

## 3. (O)MVs in Pneumonia—Release and Cargo

Although the species of pathogens causing bacterial pneumonia are manifold, almost all strains release OMVs or MVs, respectively. In pneumonia, (O)MVs may be released by both common bacterial strains—like *S. pneumoniae* [47]—but also by less common pathogens such as *P. aeruginosa*, *L. pneumophila*, *M. tuberculosis*, *A. baumannii,* and *M. catarrhalis* [48,49,50,51,52]. These (O)MVs are increasingly recognized as essential parts of the secretome of lung pathogens that may carry a variety of cargo, which includes, but is not limited to proteins, nucleic acids, fatty acids, lipoproteins, and glycolipids [47,48,49,50]. Importantly, (O)MVs can transport relevant virulence factors such as pneumolysin in *S. pneumoniae*-secreted MVs, cystic fibrosis transductance regulator (CFTR) inhibitory factor (Cif) in *P. aeruginosa*-released OMVs, and macrophage infectivity potentiator (Mip) in *L. pneumophila*-derived OMVs [47,49,53,54,55]. While (O)MV cargo thus includes potentially harmful substances, limited data are available on the distribution of virulence factors as soluble mediators versus its encapsulated form in (O)MVs. Notably, the protein content of *L. pneumophila*-released OMVs and the composition of the respective soluble fraction of the extracellular fluid differ not only quantitatively, but also qualitatively. Specifically, the majority of virulence factors were identified in OMVs, including several proteins like Mip and flagellin uniquely present in OMVs but not detectable in the extracellular fluid [49]. However, further investigations are needed to assess the distribution of virulence factors for other pathogens, and to evaluate the relevance of this subcompartmentalization for the pathogenicity of the respective bacteria.

Interestingly, accumulating data indicate that host and environmental factors are able to modify both quantity of (O)MV release and quality of (O)MV cargo. For example, *P. aeruginosa* isolates from CF patients release OMVs bearing higher concentrations of the *P. aeruginosa* aminopeptidase (PaAP) as compared to an environmental isolate, which was demonstrated to favor OMV binding to epithelial cells, even though the underlying mechanisms of enhanced OMV-binding remain elusive [56,57]. As such, the specific microenvironment of a disorder may directly foster the development of highly virulent pathogens that release potent OMVs in the alveolar compartment. Similarly, antibiotic treatment against the colonialization with pathogens may trigger OMV release. For example, *A. baumannii* produces more than twofold higher concentrations of β-lactamase-, protease-, and other protein-loaded OMVs when treated with imipenem [58]. Bacterial β-lactamases are able to deactivate a variety of β-lactam antibiotics (e.g., imipenem), pointing towards adaptive mechanisms that bacteria can upregulate in response to extrinsic stress [58]. In line with this notion, β-lactamases in OMVs from *M. catarrhalis* were found to reduce antibiotic effects of amoxicillin on *S. pneumoniae* in vitro [52]. These observations highlight the potential of (O)MVs to modulate pharmacological substances; a point of consideration that should be taken into account in the future design of treatment regimens against bacterial lung infections and in the growing problem of emerging antibiotic resistance.

Several pathogenic bacterial species causing atypical pneumonia also evolved ingenious strategies to traverse host epithelial barriers by hijacking phagocytic host cells, i.e., alveolar macrophages. In this “Trojan Horse” mechanism *M. tuberculosis* is phagocytosed and transported across the gas–blood barrier using diapedesis of the infected macrophage [59]. Moreover, mycobacteria, e.g., *M. tuberculosis* and most other pathogenic strains of this species, and legionella, e.g., *L. pneumophilia,* exploit macrophages as their replicative niche for most of the lifecycle; a competence that constitutes a defining feature of the species’ pathogenicity [60,61]. A prerequisite for this very efficient strategy is the ability of these intracellular pathogens to release molecules that modulate the host immune response. Here, (O)MVs emerge as mediators of pathogenicity that allow bacteria to outwit, exploit, or bypass host defense mechanisms. Specifically, EVs released from mycobacteria-infected macrophages bear bacterial cargo, and in turn activate proinflammatory responses and modulate innate immune mechanisms [62,63,64,65,66,67]. Of note, recent data show that virulent mediators of *M. tuberculosis* comprising bacterial lipoglycans and lipoproteins are released from macrophages in bacterial MVs as opposed to “regular” macrophage-derived EVs [68]. So far, insight into the intracellular biogenesis and release of (O)MV in host cells is limited; yet it seems plausible that the intracellular milieu of the host cell could have a major influence on (O)MV characteristics and quantity. Taken together, environmental circumstances and the intra- and extracellular host milieu have a considerable influence on bacterial (O)MV cargo and release. Therefore, these important confounders should be recognized in a comparison of the modulating effects of (O)MVs of diverse bacterial origin on the infected tissue.

## 4. The Interaction of (O)MVs with The Respiratory Epithelium—A First Step in Immunoactivation

Upon colonialization of the mucosal surfaces of bronchi and bronchioles, and/or the alveolar compartment with bacterial pathogens the epithelial barrier forms the primary antimicrobial barrier but concomitantly also the first surface of interaction with (O)MVs. (O)MVs bind to the bronchial epithelium, as shown for *P. aeruginosa* OMVs [69], and alveolar epithelial cells, as demonstrated for *A. baumannii* OMVs and *S. pneumoniae* MVs [51,70]. Notably, for EVs an important role of CD44 in the EV binding process has been well documented [71,72]. As CD44 is the hyaluronic receptor, and hyaluronan forms an important constituent of both the alveolar epithelial glycocalyx and airway mucus, a similar “attachment chemistry” may also exist in (O)MVs but remains to be identified [73,74]. Interestingly, the (O)MV–epithelial interaction is not limited to membrane binding, since *P. aeruginosa* OMVs are able to fuse with bronchial epithelial cells [75], and (O)MVs from several bacteria such as *A. baumannii* and *S. pneumoniae* can be incorporated by alveolar epithelial cells [51,70,76]. In this context it is worth to speculate whether (O)MVs could actually be transcytosed through the alveolar barrier allowing for infectious dissemination throughout the body. While such mechanisms were not described in the lung yet, it was already shown in the gut that (O)MVs are indeed able to cross epithelial barriers and enter the vascular system [77,78]. On the functional level, OMVs from a variety of lung pathogens including *P. aeruginosa*, *L. pneumophila, A. baumannii*, *K. pneumoniae*, *M. catarrhalis,* and *S. maltophilia* can induce the release of proinflammatory mediators from the epithelium. The generated cytokines include but are not limited to interleukin (IL)-1β, IL-6, IL-7, IL-8, IL-13, tumor necrosis factor α (TNF-α), interferon-γ (IFNγ), granulocyte colony-stimulating factor (G-CSF), and monocyte chemoattractant protein 1 (MCP-1) [49,56,79,80,81,82]. While these findings document a considerable range of cytokines released from the epithelium upon the OMV challenge, the underlying mechanisms how OMVs trigger cytokine release from epithelial cells remain largely unknown. That notwithstanding, the resulting epithelial cytokine responses do have a major impact on the immune response to pathogens and, importantly, also on alveolar barrier function. *P. aeruginosa* OMV challenge was shown to be sufficient to induce alveolar barrier failure, as indicated by cellular infiltration and protein leak into the alveolar space [83]. Other studies also revealed cytotoxic effects of OMVs from *L. pneumophila*, *A. baumannii*, and *S. maltophilia* on the alveolar epithelium, as indicated by epithelial delamination, mitochondrial fragmentation, and the development of a necrotic phenotype [51,82,84]. In the case of *A. baumannii* OMVs may carry the outer membrane protein A (Omp_Ab_), which activates the host GTPase dynamin-related protein 1 (DRP1) in alveolar epithelial cells in vitro [51]. DRP1 activation leads to mitochondrial fragmentation, production of reactive oxygen species, and, eventually, epithelial cell death. Both bacterial/OMV-transported Omp_Ab_ and its activation of DRP1 were shown to play important roles in the virulence of *A. baumannii* as bacterial loss of Omp_Ab_ reduced bacterial growth and systemic spread in vivo, and DRP1 RNA interference prevented OMV-induced epithelial damage [51]. Whereas this recently identified mechanism of OMV-induced mitochondrial dysfunction sparks efforts to understand OMV-mediated epithelial cell injury, further studies are needed to understand whether this pathway may constitute a common pathogenic mechanism shared by other pulmonary pathogens in the induction of alveolar barrier disruption as hallmark of respiratory failure.

Pulmonary epithelial dysfunction plays a particularly important role in CF. Affected patients suffer from genetically determined dysfunction of the anion channel cystic fibrosis transmembrane conductance regulator (CFTR), which causes decreased chloride and bicarbonate secretion from the epithelium, resulting in airway surface dehydration and impaired mucociliary clearance [16,85]. The resulting highly viscous mucoid plaques within the airways form a fertile breeding ground for infections, which are responsible for a high proportion of CF morbidity and are commonly caused by methicillin-resistant *S. aureus,* and opportunistic respiratory pathogens such as *P. aeruginosa* [17,86]. CFTR mutations are classified into six groups, with class I to III comprising no residual CFTR function, whereas class IV to VI maintain residual CFTR function, which is highly predictive for disease outcome [16]. Accordingly, inhibitory effects on residual CFTR should be avoided as they further worsen the clinical outcome in CF patients. *P. aeruginosa*—being one of the main infectious agents in CF—releases OMVs that bind to airway epithelial cells and interact specifically with cholesterol-rich lipid rafts and the neural Wiskott–Aldrich syndrome protein (N-WASP), which mediates the interaction of extracellular ligands with the actin cytoskeleton [69,75]. *P. aeruginosa*-secreted OMVs transport various virulence factors, i.e., β-lactamases, hemolytic phospholipase C, and, as aforementioned, Cif [75]. As such, OMVs can inhibit epithelial CFTR in a Cif-dependent manner [75]. Specifically, Cif inhibits the deubiquinating enzyme ubiquitin specific peptidase 10 (USP10), thereby reducing the USP10-mediated deubiquination of CFTR and promoting CFTR trafficking to and degradation in lysosomes [87]. As a result, Cif inhibits physiological cellular functions that depend on intact CFTR function [88]. Reduced USP10 activity in response to Cif also increases degradation of the transporter associated with antigen processing 1 (TAP1), lowering antigen presentation on major histocompatibility complex-I (MHC-I) molecules on epithelial cells through decreased peptide antigen translocation into the endoplasmic reticulum, which subsequently reduces the adaptive immune response [89]. Hence, *P. aeruginosa* OMVs may initiate a vicious cycle of opportunistic infection resulting in a worsening of mucociliary clearance and restriction of a competent immune response, which further increase the susceptibility to bacterial and other infections. An initial approach to overcome this pathophysiology by reducing membrane cholesterol by cyclodextrins (hydroxy-propyl-β-cyclodextrin and methyl-β-cyclodextrin) proved efficient in limiting the binding of *P. aeruginosa*-released OMVs to epithelial lipid rafts and thus restoring Cl^−^ secretion in airway epithelial cells in vitro [69]. Future research will have to probe whether strategies to reduce membrane cholesterol levels in vivo by, e.g., statins or dietary restrictions may similarly reduce OMV-binding to epithelial target cells and thus overall bacterial pathogenicity.

## 5. The Effect of (O)MVs on Innate Immunity—Novel Regulators of Immune Response

Alongside the interaction of (O)MVs with the lung epithelium their role in facilitating a proinflammatory response that activates innate immune cells has received considerable attention (Figure 1). Binding and uptake of (O)MVs was also shown for cells of the innate immune system. In ex vivo experiments, *L. pneumophila* OMVs primarily bind to macrophages within human tissue sections [84] and are subsequently internalized in a predominantly phagocytosis-independent manner [90]. Similarly, MVs from *S. pneumoniae* can be incorporated into both macrophages and dendritic cells (DCs), with uptake into DCs happening noticeably faster and again at least partially independent of phagocytosis [70,76]. (O)MVs induce inflammatory activation of innate immune cells, as reflected by increased release of cytokines including IL-1β, IL-6, IL-8, TNF-α, and CXCL2 from macrophages exposed to (O)MVs from *P. aeruginosa, L. pneumophila, H. influenzae, S. pneumonia, A. baumannii,* or *M. catarrhalis* [90,91,92,93,94,95]. In DCs, MVs from *S. pneumoniae* were likewise shown to increase the production of IL-6, IL-8, IL-10, and TNF-α, even though associated functional consequences on DC activity with respect to phagocytosis, migration to lymph nodes, and expression of costimulatory molecules remain unclear [70,76]. Analysis of underlying signaling mechanisms revealed that the stimulation of innate immune cells by (O)MVs is mediated via unique activation pathways. For example, *P. aeruginosa*-secreted OMVs trigger NLRP3 inflammasome activation, resulting in macrophage activation and IL-1β production and release [91,92]. This macrophage activation seems to be essentially dependent on toll-like receptor (TLRs)-signaling that leads to non-canonical inflammasome-dependent caspase-11 activation, which neglects the canonical AIM2 and NLRC4 inflammasome challenge [92]. Murine caspase-11 has two human homologues, caspase-4 and caspase-5, of which caspase-5 has been identified to be specifically activated by the OMV interaction with macrophages, whereas cytoplasmic LPS results in caspase-4 activation [92]. Since LPS is expected to be present on all OMVs one could speculate that LPS also accounts for TLR-dependent caspase-5 activation upon challenge with *P. aeruginosa*-derived OMVs. Indeed, evidence was provided that inhibition of LPS on *P. aeruginosa* OMVs markedly reduced macrophage activation, as indicated by decreased IL-6, TNF-α and CXCL2 release [93]. Nonetheless, further studies will have to consolidate whether macrophage activation by OMVs via the non-canonical inflammasome pathway is in fact LPS-dependent, and whether this potential mechanism can be translated to other pathogens.

Similar to *P. aeruginosa*, OMVs released from *L. pneumophila* and *A. baumannii* also activate macrophages in a TLR-dependent manner [95,96]. A proinflammatory response in uninfected macrophages can be initiated in a TLR- and myeloid differentiation factor (MyD88)-dependent way. In the case of *L. pneumophila* TLR2 is required for bystander macrophage polarization by OMVs carrying pathogen-associated molecular patterns (PAMPs) [96], which goes in line with the observation that OMVs quantitatively enhance proinflammatory cytokine secretion from initially classically activated macrophages [97]. While this activation results at first in a suppressed bacterial expansion, OMVs subsequently promote bacterial growth, as demonstrated for intracellular *L. pneumophila* replication by miRNA-146a-dependent Interleukin-1 receptor-associated kinase 1 (IRAK1) degradation within macrophages later during infection [97]. Additionally, *L. pneumophila*-secreted OMVs can attenuate phagosome–lysosome fusion in macrophages, thus further contributing to the elevated bacterial load [98].

In case of *A. baumannii* infection the induction of IL-6 secretion from macrophages depends on TLR4 signaling [95]. TLR-dependency of the innate immune response is also evident in vivo, as macrophage and neutrophil infiltration into the alveolar space in response to OMV exposure was reduced in TLR4-deficient mice [95]. Moreover, *A. baumannii*-derived OMVs also cause mitochondrial damage via an OmpA_Ab_-dependent pathway in alveolar macrophages in vivo, as mechanistically described above concerning pulmonary epithelial cells [51]. Hence, OMVs not only induce an innate immune response via various mechanisms but also promote bacterial immune evasion. Future research will need to probe to which extent either of these pathways contributes to the pulmonary congestion and neutrophil infiltration observed upon stimulus with *A. baumannii*-derived OMV [95].

While the aforementioned studies mainly focused on the cellular aspect of innate immunity and resulting changes in cytokine release, limited data are available with regard to the direct interaction of (O)MVs with humoral mediators. *S. pneumoniae*-released MVs were shown to inhibit the release of neutrophil extracellular traps (NETs) by transferring the bacterial DNAse TatD to neutrophils [55]. NETs trap extracellular bacteria and fungi in complexes of nuclear chromatin and bactericidal proteins to reduce their dissemination and limit the spread of infection [99]. Consequently, depletion of TatD in *S. pneumoniae* decreased bacterial replication in a murine pneumonia model in vivo, and as such protected against fatal outcome [55]. Another study demonstrated that *S. pneumoniae*-secreted MVs bind to the complement proteins C3, C5b-9 and factor H, and reduce complement-dependent opsonophagocytic killing of *S. pneumoniae* by macrophages in an adhesion and phagocytosis assay in vitro [70]. C3 and C5b-9 have key roles in the opsonization of pathogens [100,101], whereas factor H is considered as an inhibitor of the complement system, maintaining balance between pathogen defense and inhibiting complement activation by host factors [102]. Therefore, the exact mechanisms by which the imbalance in the complement cascade resulting from parallel binding of MVs to both stimulating and suppressing components diminishes bacterial killing remain to be elucidated.

An increasing body of work sheds light on important immunomodulatory interactions of (O)MVs with cellular and humoral components of the innate immune system in pneumonia. (O)MVs released by a variety of pulmonary pathogen species are capable of inducing potent (pro)inflammatory responses and in parallel facilitate a broad range of immune evasion mechanisms of the host pathogen, which can aggravate the course of disease. (O)MVs alone are competent to evoke pathologies comparable to bacterial pneumonia, as exemplarily shown for *K. pneumoniae*-derived OMVs [80]. Yet, the molecular mechanisms by which (O)MVs mediate evasion of pathogens from the complement system and NETs remain unclear so far and form an important topic for antimicrobial research.

## 6. (O)MVs and Adaptive Immunity

Knowledge on the effect of (O)MVs on the adaptive immune system is limited. However, the altered cytokine response of pulmonary epithelial cells and tissue-resident and immigrating innate immune cells can be expected to trigger activation of both B- and T-cells, their maturation in lung draining lymph nodes and at the site of infection, and eventually the formation of tertiary lymphoid organs in the lung. However, the evidence for lymphocyte infiltration and expansion upon stimulation with (O)MV remains limited for pneumonia. The first indication of a putative modulation of adaptive immunity by (O)MVs was provided by the observation that intracellular MVs produced by *M. tuberculosis* in infected macrophages shuttle mycobacterial lipoglycans—known as virulence factors—to CD4^+^ T-cells [50]. As a result, MVs decreased IL-2 secretion and, thereby, limited autocrine-activation of T-helper cell expansion and maturation [50]. These initial signs of T-cell anergy are highlighted by mycobacterial MVs inducing expression of the gene related to anergy in lymphocytes (GRAIL) and consequently reduced proliferation upon restimulation [50].

*M. catarrhalis* is a respiratory-tract commensal organism, which not only causes infections of the lower but also the upper respiratory tract [103]. As commensal organism *M. catarrhalis* often resides in the tonsils that are part of the pharyngeal lymphoid tissue. In fact, tonsils are secondary lymphoid organs that are constantly exposed to airborne antigens and are predominantly populated by B-cells, which accordingly play an important role in the regulation of *M. catarrhalis* infection [104]. Like other secondary lymphoid tissues, tonsils are sites of increased expansion of B-cell diversity and memory providing an environment for plasma cell differentiation and effector B-cell development, including IL-10 and IL-35 secreting regulatory B-cells (B_reg_), which play significant roles in the control of the tonsillar commensal bacterial flora [105,106]. OMVs released by *M. catarrhalis* bind to immunoglobulin D (IgD) B-cell receptors depending on moraxella IgD binding protein (MID), induce Ca^2+^ influx, and, eventually, B-cell receptor internalization [104]. On top of that, OMVs activate an inflammatory response in B-cells as indicated by IL-6 and IL-10 release and antibody production, which is attenuated in absence of MID on OMVs [104]. However, the functional consequences of this response remain unclear. While Vidakovics et al. hypothesize that this pathway leads to a T-cell-independent B-cell response with impaired bacterial killing, evidence for increased bacterial survival is outstanding and remains to be tested in further studies [104].

## 7. (O)MVs as Novel Vaccine Candidates against Pulmonary Infection

Several bacterial pathogens that induce infections in the lung are at present still poorly treatable and have detrimental long-term effects that could be prevented by effective vaccination. These include, e.g., *B. pertussis*, *S. pneumoniae*, *A. baumannii,* and *K. pneumoniae*, the two latter causing mainly nosocomial infections [107,108], whereas *B. pertussis* and *S. pneumoniae* are commonly community-acquired [109,110]. (O)MVs have been considered as auspicious vaccine candidates since they elicit potent immune responses, could be an easy-to-store off-the-shelf product, and distribute well in the organism after parenteral application [111,112]. Together, these beneficial properties initiated the development of a first clinically licensed OMV vaccine, which acts against *Neisseria meningitidis* serogroup B, representing the first effective vaccine against this meningococcal disease causing pathogen [113]. Nonetheless, like other vaccine candidates, (O)MV vaccines require careful consideration regarding the balance of immunizing effects and potential side effects, including a strong inflammatory response or even proinfectious effects in case of bacterial superinfection.

Due to the fact that *B. pertussis* has developed high levels of resistance against the acellular vaccine that is currently used in large parts of the world [114] substantial efforts have been made to develop *B. pertussis* OMVs as a vaccine candidate against whooping cough. Several studies show that application of *B. pertussis* OMVs can prevent subsequent fatal infections with *B. pertussis* and reduce bacterial load in mice [115,116,117,118], while also efficiently preventing infection by serotypes commonly unaffected by conventional acellular vaccination [119,120]. Mechanistically, OMVs induced a combined T- and B-cell response (Table 2), as demonstrated by a profound T_h1_, T_h2_, and T_h17_ response, and production of neutralizing antibodies and colonization of lung resident B- and T-cells [115,116,118,119,120,121,122]. Importantly, effective immunization by OMVs seems to depend on the presence of virulence proteins exposed on the vesicular surface, including pertussis toxin [123,124]. Accordingly, thorough testing for the safety of vaccine candidates is mandatory to exclude that OMV-associated toxins could confer harmful effects. Approaches to reduce OMV toxicity by expressing the lipid A deacylase PagL in OMVs released by *B. pertussis* yielded promising results, in that vaccine toxicity was reduced yet immunogenicity preserved [125]. Besides considerations for safety, immunogenicity, and protective efficacy, the ideal administration route and/or method for OMV vaccines remains to be determined. Indeed, intranasal inhalation as compared to subcutaneous or intraperitoneal injection of OMVs elicits qualitatively distinct immune responses, resulting in incomplete protection following subcutaneous injection as compared to intranasal application [115,116].

*S. pneumoniae* is the leading causative pathogen for CAP and, therefore, not only accounts for a high proportion of infectious mortality, but also confers serious long-term morbidity, i.e., a high incidence of cardiovascular events [10,27,29]. Three conventional vaccines are currently licensed against *S. pneumoniae* infection, which successfully reduced all-cause pneumonia and also invasive pneumococcal disease, although it remains to be shown whether all-cause mortality can be similarly reduced by these vaccines [21]. Consequently, and due to a need for broader serotype coverage and more effective long-term protection [139], efforts have been launched to test the potential of an (O)MV-based vaccine against *S. pneumoniae* infection. An initial study demonstrated the efficacy of native *S. pneumoniae* MVs applied intranasally in a mouse model, as displayed by robust IgG responses and markedly reduced mortality after infection [47]. The MVs used for these experiments were carrying well-known virulence proteins, including pneumolysin and pneumococcal surface protein A (PspA). These virulence factors were selected to induce a potent and broad immune response, as, e.g., pneumolysin is known to be present on virtually all *S. pneumoniae* serotypes [140,141]. However, it is questionable whether such vaccine formulations may be applied to humans due to considerable safety concerns, as native *S. pneumoniae* OMVs are able to induce harmful effects by interacting with humoral components of innate immunity, as described above [55,70]. Thus, a number of studies focused on OMVs from either *S. typhimurium* or *E. coli*, which were genetically modified in order to express pneumococcal proteins. Intranasal application of PspA-loaded *S. typhimurium*-derived OMVs reduced bacterial replication following subsequent *S. pneumonia* infection in vivo resulting in a considerable reduction in infection-attributed deaths [126,127]. Nevertheless, tolerability and safety concerns will also require careful assessment in this strategy, as some of the animals exhibited serious weight loss of more than 20% after vaccination [127]. With respect to safety concerns, the use of OMVs from a non-pathogenic *E. coli* strain (e.g., CLM37, which lacks O antigen and enterobacterial common antigen (ECA) synthesis [142]) may provide a promising approach. Intraperitoneally injected CLM37 *E. coli*-derived OMVs engineered to express a capsule glycan of *S. pneumoniae* serotype 14 were sufficient to induce the production of neutralizing antibodies [128]. Even though the surface molecule used in this study would not provide protection against other serotypes, it provides an encouraging proof-of-principle that glyco-engineered OMVs from non-pathogenic bacteria could present an efficient and safe vaccine strategy.

While of limited importance in community-acquired infections, *A. baumannii* has become a global threat as nosocomial pathogen accounting for both respirator-associated and systemic infections, with an emerging concern regarding multidrug-resistant strains [143]. Accordingly, development of an efficient vaccine is of great demand. Similar to vaccine development programs against *B. pertussis* and *S. pneumoniae,* native OMVs of the pathogen itself, OMVs from modified *A. baumannii* strains, and engineered OMVs from other bacteria have been tested as vaccine candidates. Native OMVs, applied mostly via the intramuscular, but also the intraperitoneal or the intranasal route, are able to induce a robust antibody response and prevent death following infection with *A. baumannii*, including multidrug-resistant strains [129,130,131,132,133]. At the cellular level, OMVs not only activated B-cells, but also upregulated the costimulatory molecules CD80 and CD86 and MHC-II in DCs in vitro, and consequently induced a robust T_h2_ response in vivo [134]. Efforts to achieve better tolerability include testing of LPS-depleted OMVs, which yielded mixed results. OMVs secreted by an *A. baumannii* strain with reduced LPS endotoxicity protected from infection-associated mortality (including multidrug-resistant strains) when administrated together with an *A. baumannii* surface protein (biofilm-associated protein (Bap) _(1-487aa)_) [135]. However, OMVs from an LPS-deficient strain (IB010) were inferior to OMVs from a wildtype strain (ATCC 19606), expressing higher levels of LPS [131]. Similar to strategies against *S. pneumoniae* discussed above, it seems promising to evaluate the immunogenic potential of OMVs from other non-pathogenic bacterial strains, which could be genetically modified to express *A. baumannii* antigens. Indeed, *E. coli*-derived OMVs genetically engineered to express the *Acinetobacter* surface protein Omp22 efficiently reduced systemic inflammatory responses and bacterial replication upon *A. baumannii* infection, and protected from a fatal course of the disease after subcutaneous injection [136]. Further studies will have to confirm whether this promising vaccine candidate can become a contender for clinical testing of new vaccination strategies against *A. baumannii.*

Quite similar to *A. baumannii, K. pneumoniae* is frequently the cause for antibiotic-resistant nosocomial infections, even though the pathogen is most prevalent in immunocompromised individuals [108]. The concept of OMV vaccination also seems to work for this pathogen as demonstrated by induction of a T_h1_ response and prevention of infectious death following intraperitoneal administration of native *K. pneumoniae*-secreted OMVs in a mouse model [137]. A novel approach evaluated whether binding of OMVs to albumin-based nanoparticles could enable an effective vaccination. A previous study already highlighted that a similar approach of coating gold nanoparticles with *E. coli* membranes allowed for efficient vaccination [144]. Indeed, the application of a nanoparticle-bound OMV vaccine was effective in preventing mice from infection with carbapenem-resistant *K. pneumoniae* after subcutaneous OMV application [138]. Intriguingly, this formulation led to even higher levels of antibodies and was more efficient in preventing fatal outcome as compared to native *K. pneumoniae*-derived OMVs, which was attributed to a potentially higher stability of the OMVs when bound to nanoparticles [138]. Taken together, (O)MVs from less pathogenic serotypes or surface-engineered (O)MVs from other, less pathogenic bacteria may pave the way for the development of not only effective but also safe (O)MV-based vaccines for airborne pathogens that continue to pose serious global threats at the current state.

## 8. Conclusions

A growing body of work is starting to highlight the role of (O)MVs in bacterial infectious diseases of the lung. While (O)MVs from various bacteria induce an inflammatory activation of both the pulmonary epithelium and innate immune cells, distinct virulence mechanisms exist for specific pathogens. Specifically, emerging evidence suggests that OMVs play an important role in the pathogenicity of *P. aeruginosa* in CF patients where they may aggravate bronchial epithelial hyposecretion in infectious exacerbations. Further effects with pathogenic potential include the induction of mitochondrial damage in both pulmonary epithelial cells and macrophages by *A. baumannii* OMVs, increased intracellular replication of *L. pneumophila* in macrophages following exposure to OMVs, inhibition of complement components and neutrophil extracellular traps by *S. pneumoniae* MVs, and induction of CD4^+^ T-cell anergy by *M. tuberculosis* MVs. At present, our insight into these effects is yet confined to a limited number of pathogen–host interactions, and a better in-depth understanding of the underlying molecular processes is required for the identification of specific targets for potential therapeutic approaches to treat pulmonary infections.

An important, yet so far rather unexplored aspect, which could additionally contribute to a better understanding of the development of bacterial infections and their virulence, is a more detailed exploration of the role of the commensal lung microbiome. While it is, for example, well known that the commensal microbiome plays a key role in maintaining gastrointestinal homeostasis not only by preventing infections but, e.g., also by protecting barrier integrity, information is limited with respect to its composition and physiological role in the lung [145,146]. Yet, an initial study on lung fibrosis in mice highlights not only the importance of the commensal pulmonary microbiome but also the pathogenic potential of its dysregulation and the mediating effects of related (O)MVs [147].

Notably, the effects of (O)MVs on the respiratory tract are not confined to respiratory infections, as (O)MVs are also present in indoor dust in sufficient amounts to induce inflammatory responses in vivo [148,149]. Potential pathophysiological effects of (O)MVs should, therefore, not only be considered in the context of infection immunology but rather also in the growing field of human-environment interaction.

Apart from their important role in bacterial pathogenicity, (O)MVs have gained considerable interest as a novel vaccine platform. To date no clinical trials for (O)MV-based vaccines against pulmonary pathogens have been performed. Yet, a number of studies highlight the potential of (O)MV-based vaccine candidates against *B. pertussis*, *S. pneumoniae, A. baumannii,* and *K. pneumoniae* in vivo, which could, when successfully tested in the clinics, help overcome limited vaccine coverage for certain bacterial serotypes and reduce the marked morbidity induced by nosocomial infections. However, one has to consider that a rocky road might still lie ahead before such ambition may turn into reality, as major obstacles still remain to be overcome. On the one hand, pharmacological aspects like optimized formulation and administration route need to be addressed. On the other hand, it remains to be resolved how (O)MVs may be utilized as safe vaccine candidates in spite of their ability to induce severe inflammatory phenotypes. First approaches, like the use of less pathogenic strains for the formulation of safer (O)MV-based vaccines, or surface-engineered (O)MVs from other, less pathogenic bacteria yield promising results, but remain to be translated into the clinical scenario. At present, (O)MVs remain a double-edged sword with both pathophysiological impact and immunizing potentials.

## Figures and Tables

**Figure 1 ijms-22-03858-f001:**
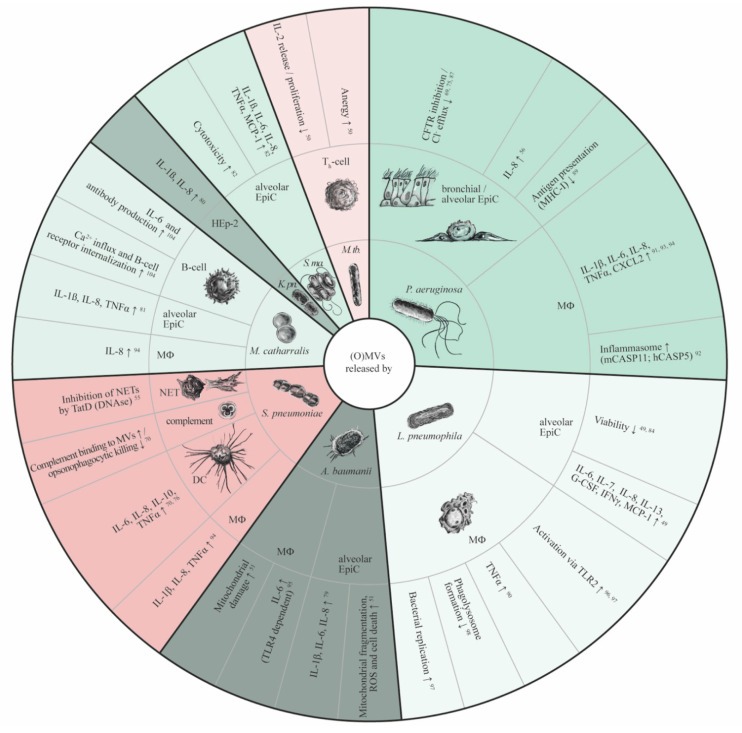
Functional properties of (O)MVs in pneumonia. Inner ring: (O)MV-releasing bacterial species; middle ring: (O)MV target cells; outer ring: (O)MV-mediated effects on target cells. Red areas: Gram-positive bacteria; green areas: Gram-negative bacteria. CFTR, cystic fibrosis transductance regulator; DC, dendritic cell; EpiC, epithelial cell; G-CSF, granulocyte colony-stimulating factor; hCASP5, human caspase-5; mCASP11, murine caspase-11; IFNγ, interferon-γ; IL, interleukin; K. pn., *Klebsiella pneumoniae*; M. tb, *Mycobacterium tuberculosis*; MCP-1, monocyte chemoattractant protein-1; MHC-I, major histocompatibility complex-I; ΜΦ, macrophage; NET, neutrophil extracellular trap; S. ma., *Stenotrophomonas maltophilia*; TLR, toll-like receptor; TNFα, tumor necrosis factor α.

**Table 1 ijms-22-03858-t001:** Microbial etiology of lower respiratory tract infections in different conditions [6,7,8,9,11,12,15,16,17,18,19,20].

Disease	Common Pathogens	Proportion *	Less Common Pathogens
CAP	*Streptococcus pneumoniae* *Haemophilus influenzae* Influenza, other respiratory viruses	13–68% 1–45% up to 71%	*Mycoplasma pneumoniae* *Legionella pneumophila* *Staphylococcus aureus* *Moraxella catarrhalis* *Klebsiella pneumoniae* *Mycobacterium tuberculosis*
HAP	*Staphylococcus aureus* *Pseudomonas aeruginosa*	15–36% 17–28%	*Klebsiella pneumoniae* *Acinetobacter baumannii* *Enterobacter* spp. *Escherichia coli* *Stenotrophomonas maltophilia* *Serratia* spp.
Predisposition			
COPD	*Haemophilus influenzae* *Streptococcus pneumoniae* *Haemophilus parainfluenzae* *Moraxella catarrhalis* Influenza, other respiratory viruses	14–39% 13–25% 13–25% 7–13% 20–40%	*Pseudomonas aeruginosa* *Chlamydia pneumoniae* *Mycoplasma pneumoniae*
CF	*Staphylococcus aureus* *Pseudomonas aeruginosa* *Stenotrophomonas maltophilia* *Haemophilus influenzae* *Achromobacter* spp.	45–80% 20–75% 10–18% 5–30% 3–30%	*Burkholderia cenocepacia* *Mycobacterium abscessus* *Mycobacterium avium-intracellulare*

* Proportion of all patients in which pathogens were identified. Underlying data were published before the SARS-CoV-2 pandemic. CAP, community-acquired pneumonia; HAP, hospital-acquired pneumonia; COPD, chronic obstructive pulmonary disease; CF, cystic fibrosis.

**Table 2 ijms-22-03858-t002:** Immunogenic properties of (O)MV vaccine candidates against bacterial pathogens causing lower respiratory tract infections.

Pathogen	Immunogenic (O)MVs	Cellular and Humoral Immunity	Vaccine Efficacy	Side Effects
*B. pertussis* 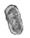	native [115,116,117,118,119,120,121,122,123,124,125]	T_h1_, T_h2_, T_h17_, T_RM_, B_RM_, plasma cells 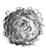 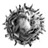 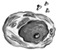	reduced bacterial load, protection against fatal outcome	weight loss
engineered (decreased endotoxicity) [125]	not reported	similar reduction in bacterial load	not reported
*S. pneumoniae* 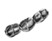	native [47]	plasma cells 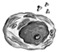	protection against fatal outcome	not reported
engineered (*S. typhimurium*/*E. coli* OMVs) [126,127,128]	plasma cells 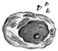	reduced bacterial load, protection against fatal outcome	weight loss (*S. typhimurium* OMVs)
*A. baumannii* 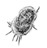	native [129,130,131,132,133,134]	T_h2_, DCs, plasma cells 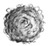 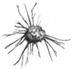 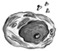	protection against fatal outcome	not reported
engineered (LPS-depletion, *E. coli* OMVs) [131,135,136]	plasma cells 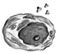	reduced inflammation and bacterial load, protection against fatal outcome	LPS-depletion: partly ineffective protection
*K. pneumoniae* 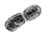	native [137]	T_h1_, plasma cells 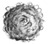 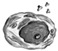	dose-dependent protection against fatal outcome	not reported
engineered (nanoparticle-bound OMVs) [138]	plasma cells 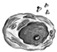	complete protection against fatal outcome	not reported

All results were reported in mouse experiments. B_RM_, resident memory B-cells; DCs, dendritic cells; LPS, lipopolysaccharide; *S. typhimurium*, *Salmonella typhimurium*; T_h1_, type-1 helper T-cells; T_h2_, type-2 helper T-cells; T_h17_, T helper-17 cells; T_RM_, resident memory T-cells.

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
