# Peer review of "Bacterial Membrane Vesicles in Pneumonia: From Mediators of Virulence to Innovative Vaccine Candidates"

_ijms, 2021, doi:10.3390/ijms22083858_

Round 1

Reviewer 1 Report

This review describes the role of Outer Membrane Vesicles and Membrane Vesicles (O)MVs produced by the species of bacteria that are most commonly associated with pneumonia in humans. The authors review the current knowledge of the (O)MVs produced by these species in terms of biogenesis, analysis, interaction with respiratory epithelia and also the immune response they generate. The authors also discuss the possibility of using both native and modified (O)MVs as vaccines. At each stage they highlight what is known but also stress that there remains a lot of work to be done in relation to (O)MVs associated with pneumonia. I found to the review to comprehensive, informative, clear and logical. I have no comments or suggestions in relation to the content. Overall, it is well written, but I think it would benefit from a finer check to improve readability.

Some minor specific comments:

Line 133- I think this should be a heading

Line 149 – “of virulence factors was identified” should be “of virulence factors were identified”

Line 195 – also should be a heading

Line 227 – Should be “In THE case of A. baumannii, OMVs may..”

Line 236 – “studies are in need to understand” should be “studies are needed to understand”

Line 239 – Cystic fibrosis has been mentioned previously

Line 318 – Should be “In THE case of L. pneumophila

Lines 382-3 – this sentence should be rewritten for clarity

Line 399 – should be a heading

Reviewer 2 Report

This manuscript review current knowledge within the field of membrane vesicles (MV) biology and possible pharmaceutical implication form bacterial species associated with pneumonia. The review is highly appropriate as much attention has been published within the field on bacterial infection associated with the gut. It also review what is known both for Gram-positive and Gram-negative bacteria.

Comments to manuscript are as follows:

Leave out the citation by William Osler. It does not add anything to the abstract.

Correct Gram-negatives and Gram-positives with capitalize G. I see that these terms are written both ways in the literature, but its origins stems from the Danish researcher Hans Christian Gram and should therefore be capitalized. 

First time a bacterial species is mentioned the full name should be stated. The name may be abbreviated from then on. Please correct this throughout the manuscript.

Line 85 – 90. There are several works that identified cytoplasmic and inner membrane associated cargo in membrane vesicles from Gram-negative bacteria. Also, transmission electron micrographs of MV isolated from Gram-negative bacteria reveals single, double and even triple membrane MV. It is speculated that the double membrane vesicles arise from bacterial lysis and that the tipple membrane vesicles could develop from encapsulated bacteriophages. Although the two latter hypotheses are not thoroughly proved they should be addressed in this part of the review.

Is line 133 a headline? Should be in bold? This also includes line 195, and 399.

Line 140 – 144. Consider if the preposition "on" should be "in" when describing the MV cargo.

Line 211. Dissemination of MVs secreted from the gut microbes through the epithelial layer and into the vascular system had been described and could mentioned here.

Line 219. The authors mention that the mechanism of cytokine release is largely unknown, yet, in the next chapter it is described in more detail. Considering revising the sentence, or clarify this statement.

Line 375. Should be mycobacterial lipoglycans, not lypoglycans?

Figure 2. it is of relevance to state which species the various vaccine trails where tested in. It might be my web browser, but the cells in the table are skewed and should be linearized. Please check the layout in different browsers before publications.

Line 502. The authors mention the nano-particle encapsulation of MVs could improve MV based vaccination safety, immune response. Such strategies have also been tested for against other bacterial diseases and could be references here.
